# Drug Carriers Based on Graphene Oxide and Hydrogel: Opportunities and Challenges in Infection Control Tested by Amoxicillin Release

**DOI:** 10.3390/ma14123182

**Published:** 2021-06-09

**Authors:** Anna Trusek, Edward Kijak

**Affiliations:** 1Group of Micro, Nano and Bioprocess Engineering, Department of Chemistry, Wroclaw University of Science and Technology, Wybrzeże Wyspiańskiego 27, 50-370 Wroclaw, Poland; 2Department of Dental Prosthetics, Wroclaw Medical University, Krakowska 26, 50-425 Wroclaw, Poland

**Keywords:** amoxicillin, bromelain, chemical activation of graphene oxide, enzymatic drug release, antibacterial carrier, *Enterococcus faecalis*, periodontal and endodontic diseases

## Abstract

Graphene oxide (GO) was proposed as an efficient carrier of antibiotics. The model drug, amoxicillin (AMOX), was attached to GO using a peptide linker (Leu-Leu-Gly). GO-AMOX was dispersed in a hydrogel to which the enzyme responsible for releasing AMOX from GO was also added. The drug molecules were released by enzymatic hydrolysis of the peptide bond in the linker. As the selected enzyme, bromelain, a plant enzyme, was used. The antibacterial nature of the carrier was determined by its ability to inhibit the growth of the *Enterococcus faecalis* strain, which is one of the bacterial species responsible for periodontal and root canal diseases. The prepared carrier contained only biocompatible substances, and the confirmation of its lack of cytotoxicity was verified based on the mouse fibrosarcoma cell line WEHI 164. The proposed type of preparation, as a universal carrier of many different antibiotic molecules, can be considered as a suitable solution in the treatment of inflammation in dentistry.

## 1. Introduction

The successful design of a drug carrier requires addressing many issues. Firstly, it is connected with a preparation of an efficient nanocarrier with optimized drug-loading capacity. The second issue is confirming or improving the carrier biocompatibility and elimination of its possible toxicity. Finally, a system able to release drugs in a controllable way with optimized dosage at a specific site required for successful therapy should be designed [1].

Graphene is a flat structure made up of carbon atoms joined together in hexagons. A theoretical description of graphene was developed as early as 1947 in a paper by Wallace [2], but the substance itself it was not produced until 2004 [3,4]. Since then, work on graphene has accelerated—both from a pure research perspective and in the search for ever better methods of producing this material. Graphene, in addtion to a large surface area, has superior mechanical, electrical, and thermal properties. Additionally, graphene can be chemical modified to produce graphene oxide (GO) and reduced graphene oxide (rGO). The presence of oxygen moieties plays a significant role in antimicrobial activity [5,6]. Additionally, the reaction of epoxy, hydroxyl, and carboxyl groups of GO with bacteria’s biomolecules can influence their cell growth and metabolic system adversely [7]. 

GO is also being investigated for its use as a drug carrier mainly to deliver anti-cancer drugs with the most commonly used agents doxorubicin and camptothecin [8], and antibodies for the selective killing of cancer cells. GO drug loading can be applied during chemotherapy and photo-thermal treatment as alone release method and in one system simultaneously [9,10]. Drugs are attached both by physical means (adsorption) and by using reactive graphene’s oxide groups [11,12,13]. Despite GO’s antibacterial properties, it has not been tested as a carrier for antibacterial drugs, including antibiotics.

Dental caries, periodontal and endodontic diseases have a close relationship with microbes. Oral microbial colonization exists in a balance in oral microenvironment. The bacterial microflora is usually composed of many species, most often of pathogenic nature. The literature on the subject is quite divergent concerning the most common bacterial cultures. The participation of individual species is strictly dependent on the primary site of infection and the therapeutic measures taken.

The first studies identified Gram-positive facultative anaerobes such as *Enterococcus faecalis* and *Streptococcus* spp. [14,15,16,17]. As shown by Rocas et al. *E. faecalis* is more associated with asymptomatic cases of primary endodontic infections than with symptomatic ones, and *E. faecalis* was found in cases of endodontic treatment failure. This bacteria was detected in 20 of 30 cases of persistent endodontic infections associated with root-filled teeth [18].

Using molecular-based detection, it was found that Gram-negative anaerobes, including *Fusobacterium nucleatum*, *Treponema denticola* and *Tannerella forsythia*, also can be responsible for periodontal disease [19,20]. Gram-negative bacteria, in particular, *F. nucleatum* elicited an enhanced pro-inflammatory response in macrophages, inhibited osteogenic differentiation and reduced cell viability [21].

Another study demonstrated that *Prevotella intermedia* was the most prevalent species of the colonies in periodontal pockets, whereas *Porphyromonas gingivalis* and *P. intermedia* were the more prevalent in root canals. Isolates of *P. gingivalis* and *P. intermedia* were simultaneously identified in root canals and periodontal pockets. Eighteen per cent of teeth exhibited the simultaneous colonization by *P. gingivalis*, *Tannerella forsythia*, and *Porphyromonas endodontalis* in the pulp and periodontal microenvironments [22]. The crucial role in primary endodontic infection of *Prevotella nigrescens* is also shown in the paper of Martinho et al. [23]. This strain was found in 57% of infections.

Such a diverse and often unknown bacterial flora requires a universal solution. Inter alia the antimicrobial properties of GO against dental pathogens were tested [24,25]. He et al. [25] used three typical bacteria of dental caries, periodontal, and periapical diseases, *S. mutans*, *P. gingivalis* and *F. nucleatum*, to evaluate the antibacterial activity of GO nanosheets in different concentrations (20, 40, and 80 μg/mL). The growth of *P. gingivalis* and *F. nucleatum* was already wholly stopped at a concentration of GO 40 μg/mL. *S. mutans* was the most resistant to GO inhibition; however, at a concentration of 80 μg/mL, it was already significantly reduced. Transmission electron microscopy (TEM) images revealed that the cell wall and membrane of tested bacteria lost their integrity, and the intracellular contents leaked out after GO treatment. Not rarely GO antibacterial activity was checked in complex with metals and metal oxides [26,27,28,29,30].

This study aimed to recognize the GO potential as a drug carrier in the treatment of inflammation in dentistry. In the paper [31], we described the efficient, utterly innovative method of the anticancer drug (doxorubicin) to GO binding via a peptide linker. A technique for covalent binding of antibiotic molecules has been developed analogously. A model drug, amoxycillin (AMOX) will be used, and as a plant-derived enzyme, bromelain (BROM) as the releasing enzyme. The antibacterial properties of the carrier will be confirmed on an *E. faecalis* strain. It is one of the bacterial species responsible for periodontal and root canal diseases. 

AMOX is a semi-synthetic β-lactam antibiotic with bactericidal activity, belonging to the aminopenicillin group (Figure 1). AMO is a broad-spectrum antibiotic. It acts on both Gram-positive (such as *Enterococcus faecalis* and *Streptococcus* spp.) and Gram-negative bacteria (such as *Prevotella* spp. and *Fusobacterium* spp.); aerobes and anaerobes. It is used to treat many infections, including dental infections. AMOX is the first-choice antibiotic for endodontic infections in both European and Asian countries [32].

BROM is a cysteine endopeptidase with broad specificity for cleavage of proteins obtained from a stem or fruit of *Ananas comosus*. 

## 2. Materials and Methods

### 2.1. Materials

The following reagents: alginic acid sodium salt, Cat. No.180947; *N*,*N*′-dicyclohexylcarbodiimide (DCC), Cat. No. D80002; divinyl sulfone, Cat. No. V3700; amoxicillin (AMOX), Cat. No. A8523; ethanolamine, Cat. No. 411000; Gly-Gly-Leu, Cat. No. G9503; N-Hhydroxysulfosuccinimide sodium salt (sulfo-NHS), Cat. No. 56485; bromelain (BROM) from pineapple stem, Cat. No. B4882; artificial saliva, Cat. No. SAE0149; 4-morpholineethanesulfonic acid, Cat. No. M3671 were from Sigma (St. Louis, MO, USA). Graphene oxide (flakes size < 20 m) from Advanced Graphene Products (Zielona Gora, Poland), cell line WEHI 164 from American Type Culture Collection (Rockville, MD, USA); *Enterobacter faecalis* PCM1861 were purchased in Polish Collection of Microorganisms PCM- PAN (Wroclaw, Poland).

### 2.2. GO-AMOX Complex Preparation 

GO with the peptide linker (Gly-Gly-Leu) was prepared according to the scheme described previously [31,33]. Divinyl sulfone is a known activator of hydroxyl groups in the literature [34,35]. Before AMOX attachment, the carboxyl groups of the linker were activated with DCC, an efficient activator of carboxyl groups [36,37]. 

The measurements of AMOX concentration were made in solutions before and after drug attachment and also in the first wash solution using high-performance liquid chromatography (HPLC) on a Waters^TM^ LC Module I plus equipped with a XTerra RP18 column (250 mm × 4.6 mm, particle size 5 m, Waters, Milford, MA, USA) and detection UV at 274 nm. As a mobile phase 0.05 M potassium phosphate, pH 5.0 (the pH adjusted to 5.0 by using potassium hydroxide) was used. The AMOX concentration was calculated according to a standard curve prepared each time before the series of measurements. Solutions with AMOX concentrations of 4, 20, 60, 180, 360 mg/L were used as standards. 

AMOX loading was visualized by FI-IR spectra (FT-IR Nicolet iS50, Thermo Scientific, Waltham, MA, USA) in the region of 400–4000 cm^−1^. Flakes size distribution before and after drug attachment was monitored using a particle analyser (Sald 2300, Shimadzu, Kyoto, Japan).

### 2.3. GO-AMOX Encapsulation in Hydrogel

The capsules were prepared of 1.8% (*w*/*v*) sodium alginate in 0.1 M MES buffer (pH 6.6). The mixture of GO flakes with attached AMOX molecules and sodium alginate solution with BROM at the concentration 0.2 mg/mL was dropped into a crosslinking bath consisting of 8% (*w*/*v*) sodium chloride solution in 0.1 M MES buffer (pH 6.6). In order to avoid protein diffusion during capsule formation, the crosslinking bath also contained BROM at the concentration 0.2 mg/mL. 

As it was presented before [38], capsules containing GO flakes were stable when the ratio of GO flakes to alginate did not exceed 1.46:1 m/v [mg/mL]. When a larger mass of flakes was used, the capsules disintegrated after about 1–2 h. Thus, the prepared capsules were prepared at the ratio of GO flakes to alginate solution 1.12:1 [mg/mL]. 

The instilled mixture to crosslinked bath solution volume ratio was 1:1.5. The crosslinked bath solution was stirred at 230 RPM. The capsules were crosslinked at 6 °C for 24 h. Then they were washed twice with 0.1 M MES buffer (pH 6.6) and stored at 6 °C in this buffer.

### 2.4. Physical Stability of GO-AMOX Alginate Capsules

The stability of the capsules was determined based on the change of their diameter after incubation in a given solution. The alginate capsules loaded with GO flakes were prepared with a 1.8% (*w*/*v*) sodium alginate in a 0.1 M MES buffer (pH 6.6). The ratio of GO flakes to alginate solution was 1.12:1 [mg/mL]. GO flakes were evenly dispersed into solution using ultrasound (power 320 W, frequency 35 kHz, time 30 min; Sonorex RK 100 H, Bandelin, Germany). 

For measuring the capsules diameter, the capsules photos and computer program, Jens Rüdigs Makroaufmaβ-programm 0.9.2 (Freiburg, Germany), were used. In each tested solution, 50 capsules having the same initial size were incubated in plastic tubes. Ten capsules as a representative sample used to take photos using a Nikon D750 camera (Chiyoda, Japan). The capsules’ photos were taken every two days by 30 days. The stability of the capsules was checked in 0.1 M MES buffer (pH 6.6), in artificial saliva (pH 6.8) and demineralised water (pH 6.8) in 37 °C. 

### 2.5. Enzymatic Hydrolysis—Antimicrobial Properties of GO-AMOX Alginate Capsules

GO-AMOX complex (1 mg) was suspended in 2 mL of 1.8% (*w*/*v*) sodium alginate in 0.1 M MES buffer (pH 6.6) with BROM at concentration in the range of 0.04–0.4 mg/mL. The mixture of GO-AMOX and BROM solution was dropped into a crosslinking bath—8% (*w*/*v*) sodium chloride solution in 0.1 M MES buffer (pH 6.6). To avoid protein diffusion during capsule formation, the crosslinking bath contained BROM at the same concentration as the instilled mixture. The crosslinked bath solution was stirred at 230 RPM. The capsules were crosslinked at 4 °C for 24 h. Then they were washed twice with 0.1 M MES buffer (pH 6.6) and the release of AMOX was monitored by 24 h using HPLC. AMOX concentration was determined according the method described in the Section 2.2. 

Figure 2 shows the scheme of described drug carrier preparation. 

The antibacterial properties of the carrier were tested in a reductive culture of *E.faecalis* was performed on sterile nutrient agar plates. The plates (in 2 replicates) were lined with alginate capsules containing GO-AMOX (1:2 mg_GO_/mL_Alginate_, m_AMOX_/m_GO_ = 0.18) and BROM (0.2 mg/mL) and alginate capsules containing dissolved AMOX at concentration 0.1 mg/mL_Alginate_.

### 2.6. Cytotoxic Effects of Prepared Drug Carriers 

Cytotoxic effects of an AMOX, GO, GO-AMOX complex, and GO-AMOX + BROM solution on the cell line of mice fibrosarcoma WEHI 164 were observed using propidium iodide [39,40]. Cells cultivation and viability were performed under conditions described previously [31].

## 3. Results

### 3.1. GO Flakes Characteristic 

The preparation of GO used consists of about 61.79% carbon, 37.78% oxygen. It contains carbonyl, carboxyl, hydroxyl and epoxy groups [41], what was confirmed by FT-IR images (Figure 3). The flakes size was in the range of 1–18 μm with a dominant size in the range 3–11 μm (Figure 4). 

The GO FT-IR corresponds to the images presented in the literature [42,43,44]. The GO spectrum as shown in Figure 3 shows a broad peak located at 3700–2500 cm^−1^ attributed to the stretching vibration in hydroxyl groups, signals at 1720 and 1614 cm^−1^ belonging to the carbonyl groups, signals at 1300 and 1210 cm^−1^ due to deformation vibration in C-OH, 1042 cm^−1^ represented by C-O groups sometimes designated to C-O-C groups, and peak at 966 cm^−1^ that may be assigned to the epoxy groups.

### 3.2. GO-AMOX Complex Preparation

The binding efficiencies of the peptide and AMOX were based on the mass balance of the given compound from the solutions before and after binding—Table 1. The mass of the attached peptide was 0.145 ± 0.031 mg per 1 mg of GO, and the binding efficiency was close to 100% when the dose-volume was appropriately matched to the mass of GO. 

The GO flakes containing on average 3.94 × 10^17^ molecules of the peptide linker were used for AMOX attachment. Independently from the AMOX solution volume, the bound mass was 0.1824 ± 0.026 mg per 1 mg GO, which corresponds to around 3 × 10^17^ molecules per 1 mg GO. The AMOX attachment was visualized by FT-IR spectra and particles size increase—Figure 3 and Figure 4, respectively. 

### 3.3. Physical Stability of GO-AMOX Alginate Capsules

Due to the good dispersion of GO flakes using ultrasound the obtained capsules were pseudo-homogeneous (contained uniform dispersion of GO flakes)—Figure 5. Capsules’ diameter was in the range of 3.735 ± 0.224 mm.

The stability of the capsules was determined based on the change of their diameter after incubation in 0.1 M MES buffer (pH 6.6), in artificial saliva (pH 6.8) and demineralised water (pH 6.8) by 30 days. The observed changes are shown in Figure 6. Over 30 days, the capsules did not disintegrate in any of the solutions. Despite the increase in capsule diameter, GO flakes did not flow out of the capsules. Also, the significant presence of monovalent cations in the artificial saliva [45] did not adversely affect the stability of the capsules.

### 3.4. Enzymatic Hydrolysis—Antimicrobial Properties of GO-AMOX Alginate Capsules

BROM, an enzyme of plant origin, was used in the study. According to the literature, the optimum activity and high stability of this enzyme occur at pH 7.0 and a temperature of 30–37 °C [46,47]. Due to the potential use of the developed drug carrier in dentistry, the studies were conducted at pH 6.6, corresponding to the normal pH of saliva, at the human body temperature of 37 °C.

The protein linker used in the attachment of the antibiotic molecules was selected to the substrate specificity of BROM. It is the enzyme with low substrate specificity; however, from the amino acid composition of peptides obtained after BROM treatment, some preferences of BROM can be found [48]. Dominate the peptides with glycine at the N’ terminal while neutral amino acids (leucine, phenylalanine, alanine) at the C’ terminal of peptide chains. Hence, the linker used was the Gly-Gly-Leu peptide.

Table 2 shows the effect of enzyme concentration on the release efficiency of AMOX from the carrier. A BROM solution of 0.2 mg/mL was selected to further research. More than 90% of bound AMOX molecules were released at this concentration within 24 h—Table 2.

The antibiotic molecules released by hydrolysis contain attached leucine. The molecular weight of the released molecules was visualised by mass spectrometry (Q-Tof Mass Spectrometer, Bruker Daltonics, Billerica, MA, USA). An amino acid blocking the NH_2_ group, as described in the literature [49,50] doesn’t influence the antimicrobial properties of AMOX. 

Figure 7 compares the growth of *E. faecalis* cultures in the presence of alginate capsules with AMOX dissolved in the carrier and the presence of capsules including GO-AMOX and BROM at the concentration 0.2 mg/mL. In both cases, inhibition of growth (zone II) was observed at the site of capsule placement. When drugs are encapsulated in an alginate network, the diffusion of drug molecules is very fast [51,52]. Hence, the effect of such capsules was obtained as expected. However, studies have shown that also molecules bound to GO prevent strain growth. This fact confirms that BROM encapsulated inside the carrier effectively releases AMOX. The exemption of AMOX from the GO was also made visible through FT-IR and particle size analysis. After 24 h of enzyme activity, the GO particle size is similar to that before AMOX attachment (Figure 4). In the FT-IR image, there is a visible loss of signal at 2920 cm^−1^, which is visible in the image for AMOX and appeared in the GO-AMOX image (Figure 3). 

### 3.5. Cytotoxic Effects of Prepared Drug Carries

Although the cytotoxic effect of GO flakes has been investigated previously [31], cytotoxicity of the prepared carriers with AMOX was determined—Figure 8. 

The cells cultured in the absence of additions served as a control served, for which a value of 94.1% of viable cells was obtained. In the presence of GO flakes and GO with bound antibiotic, the values obtained are only slightly lower (>90% live cells), which confirmed the lack of GO cytotoxicity. The presence of antibiotic improves a little bit of cell viability, both for AMOX solution and AMOX enzymatically released from the carrier. Such effect as for AMOX was reported in the literature for other antibiotics [53,54].

## 4. Discussion

In the paper of Tahriria et al. [55] about the opportunities and challenges of graphene and its derivatives in dentistry, it was presented that graphene–based materials can be used to improve characteristics of dental materials. For example, the addition of graphene or its derivatives, e.g., to resins, cement can improve their mechanical properties, increase surface area, and enhance their bioactivity. GO coating of collagen membranes promote the process of osteoblastic differentiation, and decrease inflammation [56]. Graphene nanoplates are used as a nanofiller in a commercial dental adhesive to combat bacterial growth susceptibility [57]. Following the recommendation about the local treatment of infection in dentistry [58], the new challenges in GO addition were demonstrated in this research. 

The proposed kind of carriers was based on GO and hydrogel as an environment-friendly enzymatic activity [59]. The combination of GO with hydrogels was described previously in another drug delivery context. GO-based hydrogels present functional properties, for instance, pH-responsiveness, good mechanical properties, and thermal stabilities [60]. Thus this combination was tested, i.e., in curing bacterial infections in the gastrointestinal tract [61,62].

The presented role of BROM encapsulated together with GO-AMOX complex was to hydrolyse the peptide bond between the AMOX molecule and a peptide linker (Gly-Gly-Leu) selected to match the substrate specificity of the enzyme. Once released from the carrier, the AMOX molecule readily diffused through the hydrogel network and could penetrate the site of action (infected area). As it was shown, the enzyme activity (concentration inside hydrogel) control the rate of drug release. From here, the dosage can be planned throughout the therapy.

For the first time, BROM was used as a catalyst applied in drug molecules release. In addition to its catalytic properties, BROM can act as a phytotherapeutic drug. Its therapeutic properties were discovered several years ago; hence the potential of this substance has not yet been fully exploited. In dentistry, BROM has been used for its anti-inflammatory action, especially after the extraction of third molars. The effect of BROM was better than that of paracetamol and similar to diclofenac and ketoprofen [63]. A recent literature review has also shown that BROM is effective in reducing inflammation and oedema. Minimum inhibitory con-centration of BROM was tested on isolated strains *of Streptococcus mutans*, *Enterococcus faecalis*, *Aggregatibacter actinomycetemcomitans*, and *Porphyromonas gingivalis*. *S. mutans* showed sensitivity at the lowest concentration of 2 mg/mL, *P. gingivalis* at 4.15 mg/mL, *A. actinomycetemcomitans* at 16.6 mg/mL, while *E. faecalis* at 31.25 mg/mL [64].

Based on the obtained data and using the antibacterial properties of BROM, an ideal combination would be developing such a drug carrier in which BROM is used firstly to release drug molecules and then leave the carrier to complete the antibacterial therapy. It is a matter of selecting a suitable outer coating covering the hydrogel [65,66].

## 5. Conclusions

A carrier with progressively released antibiotic molecules could replace, i.e., the paste currently used to treat periodontal abscesses. In addition to the antibiotic, the paste components are glycerol solvent, glycerol monostearate having emulsifying properties, and paste consistency and parabens, e.g., propyl *para*-hydroxybenzoate and methyl *para*-hydroxybenzoate having preservative properties. These are substances that can cause severe allergies [67,68].

The study showed that the enzyme encapsulated together with flake GO to which antibiotic molecules are chemically attached releases drug molecules, affecting the inhibition growth of bacteria sensitive to the antibiotic. Unlike pastes, in which the mass of the administered drug is usually used in excess, the rate of drug release is controlled by the enzyme concentration chosen.

The procedure of carrier preparation used does not exclude the attachment of two different drugs simultaneously. Thus, our further research will focus on the co-immobilisations of two different antibiotics to which a wide range of bacterial strains will show sensitivity. In this way, universal drug carriers for periodontal and endodontic diseases treatment will be obtained.

## Figures and Tables

**Figure 1 materials-14-03182-f001:**
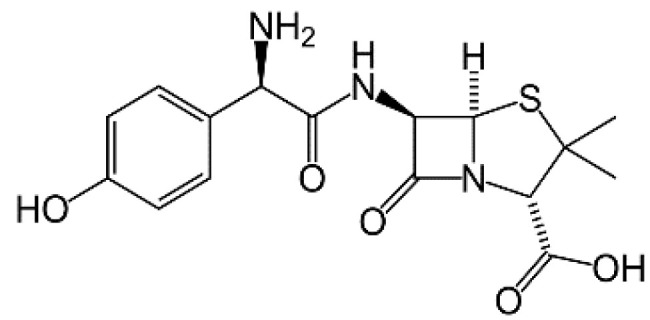
The structure of amoxycillin (AMOX).

**Figure 2 materials-14-03182-f002:**
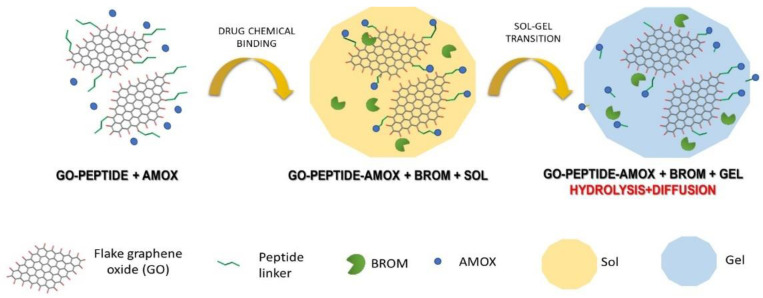
The scheme of preparing a drug carrier based on the covalent attachment of drug molecules and their enzymatic release and diffusion.

**Figure 3 materials-14-03182-f003:**
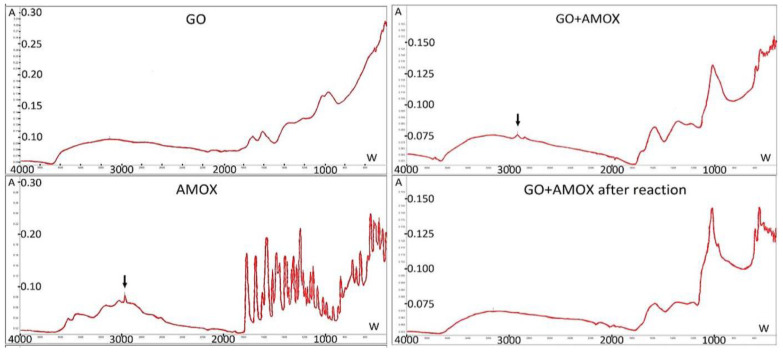
FT-IR spectra of AMOX, GO, GO with AMOX attached, and GO-AMOX after enzymatic reaction (C_E_ = 0.2 mg/mL, 72 h). A—absorbance, W—wavenumbers [cm^−1^].

**Figure 4 materials-14-03182-f004:**
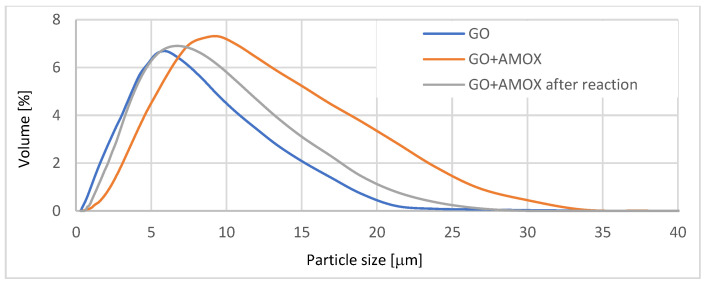
GO flakes size distribution.

**Figure 5 materials-14-03182-f005:**
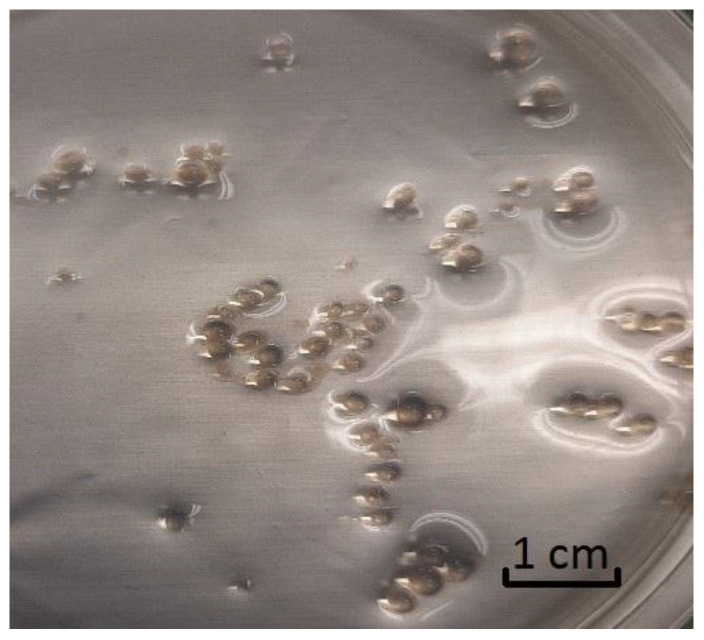
The image of GO-AMOX alginate capsules.

**Figure 6 materials-14-03182-f006:**
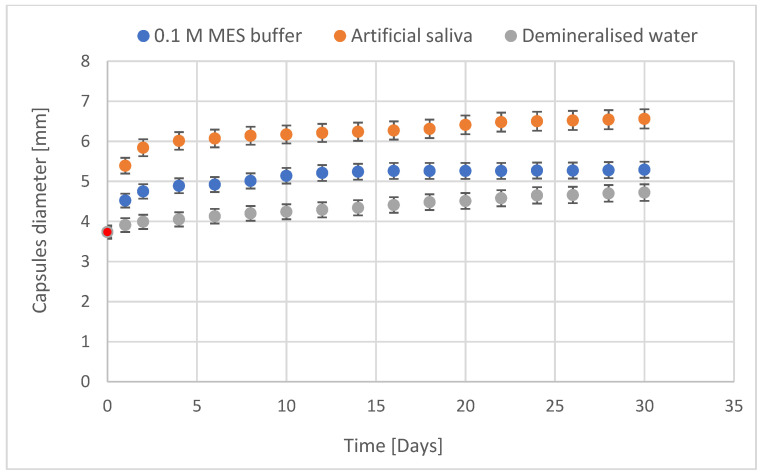
GO-AMOX alginate capsules stability expressed by their diameter (T = 37 °C). Initial capsules’ diameter was 3.735 ± 0.224 mm (a red point).

**Figure 7 materials-14-03182-f007:**
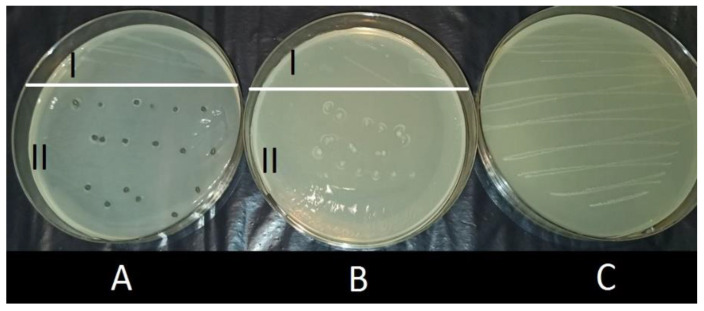
Effect of growth inhibition of *E.faecalis* by AMOX—a reduction culture (37 °C, 72 h). (**A**) GO-AMOX + BROM alginate capsules (1:2 mg_GO_/mL_Alginate_, m_AMOX_/m_GO_ = 0.18, C_E_ = 0.2 mg/mL_Alginate_); (**B**) Alginate capsules with AMOX (C_AMOX_ = 0.09 mg/mL_Alginate_); (**C**) Control.

**Figure 8 materials-14-03182-f008:**
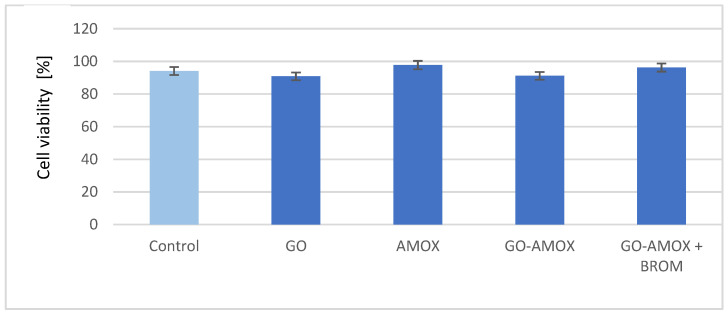
Mice fibrosarcoma WEHI 164 cells viability in the presence of GO (1 mg_GO_/mL), AMOX solution (180 μg/mL), GO-AMOX (1 mg_GO_/mL, m_AMOX_/m_GO_ = 0.18), GO-AMOX and the enzyme (BROM) solution (0.1 mg_GO_/mL, m_AMOX_/m_GO_ = 0.18, C_E_ = 0.2 mg/mL). Control—the cells growth without any additions.

**Table 1 materials-14-03182-t001:** Binding efficiency of the linker and AMOX (average values from four measurements calculated on 1 mg of GO).

	Volume [mL]	Number of Molecules Administered	Number of Molecules Attached	Efficiency [%]
Gly-Gly-Leu	13	3.19 × 10^17^ ± 5.69 × 10^16^	3.16 × 10^17^ ± 6.61 × 10^16^	99.06
	26	6.39 × 10^17^ ± 9.32 × 10^16^	3.94 × 10^17^ ± 8.27 × 10^16^	61.66
AMOX	6	4.17 × 10^17^ ± 3.13 × 10^16^	3.05 × 10^17^ ± 4.21 × 10^16^	74.39
	8	5.56 × 10^17^ ± 5.51 × 10^16^	3.14 × 10^17^ ± 4.78 × 10^16^	56.47

**Table 2 materials-14-03182-t002:** The efficiency of the AMOX releasing (average values from four measurements) with BROM encapsulated in alginate by 24 h. The initial number of AMOX molecules attached to 1 mg of GO was 3.14 × 10^17^ ± 4.78 × 10^16^.

BROM Concentration [mg/mL]	Number of AMOX Molecules Released	Efficiency [%]
0.04	0.92 × 10^17^ ± 1.17 × 10^15^	19.30
0.10	1.95 × 10^17^ ± 8.61 × 10^15^	62.10
0.20	2.84 × 10^17^ ± 1.85 × 10^16^	90.45
0.40	3.02 × 10^17^ ± 2.21 × 10^16^	96.18

## Data Availability

The data presented in this study are available on request from the corresponding author. The data are not publicly available due the continuation of patentable research.

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
