# Peer review of "Drug Carriers Based on Graphene Oxide and Hydrogel: Opportunities and Challenges in Infection Control Tested by Amoxicillin Release"

_materials, 2021, doi:10.3390/ma14123182_

Round 1

Reviewer 1 Report

The article "Drug carriers based on graphene oxide and hydrogel: opportunities and challenges in infection control tested by amoxicillin release" presents satisfactory results about the carrier prepared with Graphene oxide (GO), bromelain (BROM), amoxicillin (AMOX), only biocompatible substances with no effect on the cell line of mice fibrosarcoma WEHI 164. According to the authors, the antibacterial properties of the carrier were confirmed on Enterococcus faecalis strain and can open good perspectives to apply for treat inflammation in dentistry. I recommend publishing after a major revision by the authors.

Comments:
1.    The authors need to rewrite the confused abstract. Suggestion: show briefly work motivation, objective, methodology, results, and conclusions.

2.    Introduction: Rewrite your introduction; it became long and confused. At the first moment, my impression was I had read one review about GO. Please try to make it direct and shorter than the first version.

3.    Linea 251: Show company, city, and country of “Universal 320 R centrifuge”. Make sure that all equipment used had complete information also as you do in linea 293: “particle analyzer (Sald 2300, Shimadzu, Japan).” and others...

4.    Linea 240: 2.2. Analysis of AMOX concentration: Describe your experiment in the order it happened. In my view, the carriers were initially prepared, and only after this step the drug, AMOX, as quantified.

5.    Linea 321:  why is this sentence in bold?

The initial number of AMOX molecules attached to 1 mg of GO was 3.14.1017...

6.    Methods: I did find only 2 or 3 references that the authors can follow to do their work. My question is: Did you develop all methodology by yourself without previous citation scientific works?

7.    Linea 346 : Correct: TM-1000, Hitachi, (Japan)) ......(TM-1000, Hitachi, Japan)

8.    Linea 357: improve the quality of  Fig. 4. FT-IR spectra of AMOX, GO, GO with AMOX attached, and GO-AMOX after enzymatic reaction (CE=0.2 mg/mL, 72 h).

9.    Linea 362: Fig. 5. GO flakes size distribution measured with Sald 2300 (Shimadzu). Delete the name of Sald 2300 manufacture. 

10. Linea 386: 3.3. Physical stability of GO-AMOX alginate capsules: Linea- 387-388 The sentence “The alginate capsules were prepared of 1.8% (w/v) sodium alginate in 0.1 M MES 387 buffer (pH 6.6).” must be in the method

11. Linea 390: all equipment and specifications must appear in methods, never in results - ex. Bandelin Sonorex RK 100 H, Germany

12. Linea 392:  correct: range of 3.735-0.224 mm.

13. Linea 395: Specific which technique you used for this image (Figure 6. The image of GO-AMOX alginate capsules).

14. Linea 404: The authors must show the n value and replotted the graphic with mean and standard deviation.

15. Linea 423: ...average values from four measurements... 
Authors must show media value and standard deviation.

16.  Linea 420: Why you selected the BROM solution of 0.2 mg/mL for further research and didn't 0.40 mg/mL?

17. Linea 457: Figure 9, why did selected mice fibrosarcoma WEHI 164 cells for cell viability?

18. Linea 468: Discussion. Suppose it was possible rewriting results and discussion together. In this section, the authors did a noticeably short discussion in contrast with their introduction. My suggestion is to take some parts of the introduction and write them in the Discussion section, correlate with your obtained data.

19. Linea 486: Check in the whole text correct scientific spell of E.faecalis.

20. Linea 497: 5. Conclusions: Please show the main conclusions based on your results and make sure that you also presented your work's perspectives.

Author Response

Thank you very much for your valuable comments. The article has been reviewed and partly rewritten. 

  1. The authors need to rewrite the confused abstract. Suggestion: show briefly work motivation, objective, methodology, results, and conclusions.

The abstract has been rewritten.

  1. Introduction: Rewrite your introduction; it became long and confused. At the first moment, my impression was I had read one review about GO. Please try to make it direct and shorter than the first version.

The Introduction has been rewritten.

  1.    Linea 251: Show company, city, and country of “Universal 320 R centrifuge”. Make sure that all equipment used had complete information also as you do in linea 293: “particle analyzer (Sald 2300, Shimadzu, Japan).” and others...

Corrected in all places.

  1.    Linea 240: 2.2. Analysis of AMOX concentration: Describe your experiment in the order it happened. In my view, the carriers were initially prepared, and only after this step the drug, AMOX, as quantified.

Description of the analytical method has been added to the section ”AMOX attachment via peptide linker”.

  1.    Linea 321:  why is this sentence in bold?

The initial number of AMOX molecules attached to 1 mg of GO was 3.14.1017...

An editorial error. I apologise.

  1.    Methods: I did find only 2 or 3 references that the authors can follow to do their work. My question is: Did you develop all methodology by yourself without previous citation scientific works?

Most of the methodology was developed by the authors and partially described in earlier papers (appropriate citations were mentioned). The corrected description was shortened, giving references to previous papers.  Citations have also been added to the use of activators of hydroxyl and carboxyl groups, which are used in the developed methodology.

  1.    Linea 346 : Correct: TM-1000, Hitachi, (Japan)) ......(TM-1000, Hitachi, Japan)

Corrected.

  1.    Linea 357: improve the quality of  Fig. 4. FT-IR spectra of AMOX, GO, GO with AMOX attached, and GO-AMOX after enzymatic reaction (CE=0.2 mg/mL, 72 h).

Corrected.

  1.    Linea 362: Fig. 5. GO flakes size distribution measured with Sald 2300 (Shimadzu). Delete the name of Sald 2300 manufacture. 

Corrected.

  1. Linea 386: 3.3. Physical stability of GO-AMOX alginate capsules: Linea- 387-388 The sentence “The alginate capsules were prepared of 1.8% (w/v) sodium alginate in 0.1 M MES 387 buffer (pH 6.6).” must be in the method

Corrected

  1. Linea 390: all equipment and specifications must appear in methods, never in results - ex. Bandelin Sonorex RK 100 H, Germany

Corrected.

  1. Linea 392:  correct: range of 3.735-0.224 mm.

Corrected.

  1. Linea 395: Specific which technique you used for this image (Figure 6. The image of GO-AMOX alginate capsules).

The capsules were 3.5-4 mm in size. A Nikon camera was used to visualise them. The information was added to the paper.

  1. Linea 404: The authors must show the n value and replotted the graphic with mean and standard deviation.

The figure was replotted. The deviation was added.

  1. Linea 423: ...average values from four measurements... 
    Authors must show media value and standard deviation.

Added.

  1.  Linea 420: Why you selected the BROM solution of 0.2 mg/mL for further research and didn't 0.40 mg/mL?

The reaction efficiency for a concentration of 0.2 mg/mL was only slightly lower (<6%) than for a concentration of 0.4 mg/mL. The cost of the enzyme is one of the economic aspects.

  1. Linea 457: Figure 9, why did selected mice fibrosarcoma WEHI 164 cells for cell viability?

In addition to testing the cytotoxicity of the formulation itself (GO), the aim was also to determine whether the carrier containing the released AMOX could affect fibrosarcoma cells to some extent. Some antibiotics (such as doxorubicin) also act as cytostatics. The effect of AMOX on cancer cells is not reported in the literature.

  1. Linea 468: Discussion. Suppose it was possible rewriting results and discussion together. In this section, the authors did a noticeably short discussion in contrast with their introduction. My suggestion is to take some parts of the introduction and write them in the Discussion section, correlate with your obtained data.

Discussion has been rewritten.

  1. Linea 486: Check in the whole text correct scientific spell of E.faecalis.

Corrected.

  1. Linea 497: 5. Conclusions: Please show the main conclusions based on your results and make sure that you also presented your work's perspectives.

Conclusion part was extended.

Reviewer 2 Report

The manuscript presented by A. Trusek and E Kijak is proposing a very interesting material for dental treatment. The paper is well written and deserve to be published in "Materials" journal.

Before publishing I recommend to the authors to check the manuscript for small English errors and the units because in some places the Greek letters are not seen.

Author Response

English was carefully checked. Greek letters have been corrected, nevertheless, there is probably a problem changing them when loading an article into the system.

Reviewer 3 Report

Authors proposed a paper entitled “Drug carriers based on graphene oxide and hydrogel: opportunities and challenges in infection control tested by amoxicillin release” for the publication in Materials, MDPI.

In my opinion, this work is characterized by High scientific soundness.

A quite good use of English has been employed in this work and in the description of the results; only minor spell checks are necessary.

Due to the high number of abbreviations used in this paper, I suggest adding an abbreviation list, according to the guidelines of this Journal.

Here is the list of my observations and comments:

Abstract Line 17. Maybe better “1 mg” instead of “1 milligram”

Line 19. “a sol of sodium alginate”. maybe better to extend the term “sol” in “solution”.

Introduction Line 71. “with microenvironment in oral”. I would say “in oral microenvironment”

Line 83. “associated with root filled teeth.”. A reference could be added here.

Line 88. Are we sure that “fastidious” is the correct adjective in this context?

Line 140. “key to building”. maybe better “a key to build”

Line 163. “to improved their” better “to improve”

In my opinion, figures 1 and 2 are not totally necessary, since they just represent molecules whose molecular structure is easily findable on the web.

There is not a precise section where the aims of this paper are well declared. please add a paragraph at the end of the introduction section.

Moreover, at line 221, “The antibacterial properties of the carrier were confirmed on E. faecalis strain “. this means that a comment on results has already been introduced here. I suggest moving results and comments on the results in the proper section, and give here only the introducing information and aims of the paper.

Materials and methods. Line 251. “RPM for 30 min.” full stop not necessary since then the sentence continues.

Line 310. “on the change of their”. maybe better “on the variation of”

Line 351. “GO purchased from Advanced Graphene Products (Poland)”. This goes to materials section.

Table 1. How did you calculate efficiency of the attached molecules? do you assume efficiency and yield have the same mining in this context?

Figure 6. As this is a macroscopic image, I suggest adding a reference bar also.

Could it be possible to add errors on bars of diagram 6 ?

Line 462. “avalue” should be “a value”.

I suggest adding a short paragraphs on future perspective, in the Conclusions section.

Thank you.

Author Response

Thank you very much for your valuable comments. The article has been reviewed and partly rewritten.

Abstract Line 17. Maybe better “1 mg” instead of “1 milligram”

The abstract was rewritten.

Line 19. “a sol of sodium alginate”. maybe better to extend the term “sol” in “solution”.

The abstract was rewritten.

Introduction Line 71. “with microenvironment in oral”. I would say “in oral microenvironment”

Corrected

Line 83. “associated with root filled teeth.”. A reference could be added here.

Added.

Line 88. Are we sure that “fastidious” is the correct adjective in this context?

Corrected

Line 140. “key to building”. maybe better “a key to build”

This part was removed.

Line 163. “to improved their” better “to improve”

Corrected

In my opinion, figures 1 and 2 are not totally necessary, since they just represent molecules whose molecular structure is easily findable on the web.

The diagram of the GO structure has been removed. The AMOX diagram, for the convenience of the reader, remains. However, it can be removed in the final version if desired. 

There is not a precise section where the aims of this paper are well declared. please add a paragraph at the end of the introduction section.

The aim of the paper was declared.

Moreover, at line 221, “The antibacterial properties of the carrier were confirmed on E. faecalis strain “. this means that a comment on results has already been introduced here. I suggest moving results and comments on the results in the proper section, and give here only the introducing information and aims of the paper.

Corrected

Materials and methods. Line 251. “RPM for 30 min.” full stop not necessary since then the sentence continues.

Corrected

Line 310. “on the change of their”. maybe better “on the variation of”

Corrected

Line 351. “GO purchased from Advanced Graphene Products (Poland)”. This goes to materials section.

Removed from results.

Table 1. How did you calculate efficiency of the attached molecules? do you assume efficiency and yield have the same mining in this context?

From the mass balance (concentration determined on HPLC x volume) in solutions before and after binding. The measurement deviation was added. 

Figure 6. As this is a macroscopic image, I suggest adding a reference bar also.

A reference bar was added.

Could it be possible to add errors on bars of diagram 6 ?

Errors were added.

Line 462. “avalue” should be “a value”.

Corrected

I suggest adding a short paragraphs on future perspective, in the Conclusions section.

Future perspective was described.  

Reviewer 3:

Abstract Line 17. Maybe better “1 mg” instead of “1 milligram”

The abstract was rewritten.

Line 19. “a sol of sodium alginate”. maybe better to extend the term “sol” in “solution”.

The abstract was rewritten.

Introduction Line 71. “with microenvironment in oral”. I would say “in oral microenvironment”

Corrected

Line 83. “associated with root filled teeth.”. A reference could be added here.

Added.

Line 88. Are we sure that “fastidious” is the correct adjective in this context?

Corrected

Line 140. “key to building”. maybe better “a key to build”

This part was removed.

Line 163. “to improved their” better “to improve”

Corrected

In my opinion, figures 1 and 2 are not totally necessary, since they just represent molecules whose molecular structure is easily findable on the web.

The diagram of the GO structure has been removed. The AMOX diagram, for the convenience of the reader, remains. However, it can be removed in the final version if desired. 

There is not a precise section where the aims of this paper are well declared. please add a paragraph at the end of the introduction section.

The aim of the paper was declared.

Moreover, at line 221, “The antibacterial properties of the carrier were confirmed on E. faecalis strain “. this means that a comment on results has already been introduced here. I suggest moving results and comments on the results in the proper section, and give here only the introducing information and aims of the paper.

Corrected

Materials and methods. Line 251. “RPM for 30 min.” full stop not necessary since then the sentence continues.

Corrected

Line 310. “on the change of their”. maybe better “on the variation of”

Corrected

Line 351. “GO purchased from Advanced Graphene Products (Poland)”. This goes to materials section.

Removed from results.

Table 1. How did you calculate efficiency of the attached molecules? do you assume efficiency and yield have the same mining in this context?

From the mass balance (concentration determined on HPLC x volume) in solutions before and after binding. The measurement deviation was added. 

Figure 6. As this is a macroscopic image, I suggest adding a reference bar also.

A reference bar was added.

Could it be possible to add errors on bars of diagram 6 ?

Errors were added.

Line 462. “avalue” should be “a value”.

Corrected

I suggest adding a short paragraphs on future perspective, in the Conclusions section.

Future perspective was described.  

Round 2

Reviewer 1 Report

The article " Drug carriers based on graphene oxide and hydrogel: opportunities and challenges in infection control tested by amoxicillin release " presents satisfactory results about the carrier prepared with Graphene oxide (GO), bromelain (BROM), amoxicillin (AMOX), only biocompatible substances with no effect on the cell line of mice fibrosarcoma WEHI 164. According to the authors, the antibacterial properties of the carrier were confirmed on Enterococcus faecalis strain and can open good perspectives to apply for treat inflammation in dentistry. After a review of the authors, I recommend publishing in the present form.

Author Response

Thank you for accepting the article after correction. 

Reviewer 3 Report

Authors provided a new version of their paper, entitled  “Drug carriers based on graphene oxide and hydrogel: opportunities and challenges in infection 2 control tested by amoxicillin release”

The topic of drug carrier is very hot today, in this pandemic situation, in order to be effective with the maximal limitation of the side effects. in this regard, also the techniques employed for the production of drug carriers need to be re-thinked in order to achieve these goals.

Line 62. we cannot say that bacteria are demanding.

In my opinion, the quality of figure 3 is still very low.

Do you have the possibility to add error bars in figure 8?

In the Discussion paragraph, I think that the main sentence/verb in this paragraph is missing:

“For example, the biocomposites with 302 tunable physicochemical/biological properties that can be synthesized by functionaliza-303 tion and combination of graphene and its derivatives with other biomolecules and bio-304 materials in order to obtained specific characteristics, such as high mechanical properties, 305 large surface area as well as enhanced bioactivity.”

Author Response

Line 62. we cannot say that bacteria are demanding.

Corrected. 

In my opinion, the quality of figure 3 is still very low.

Corrected. 

Do you have the possibility to add error bars in figure 8?

Corrected. 

In the Discussion paragraph, I think that the main sentence/verb in this paragraph is missing:

“For example, the biocomposites with 302 tunable physicochemical/biological properties that can be synthesized by functionaliza-303 tion and combination of graphene and its derivatives with other biomolecules and bio-304 materials in order to obtained specific characteristics, such as high mechanical properties, 305 large surface area as well as enhanced bioactivity.”

Rewritten.